# Preoperative Antibiotic Administration Does Not Improve the Outcomes of Operated Diabetic Foot Infections [note 1]

**DOI:** 10.3390/antibiotics13121136

**Published:** 2024-11-26

**Authors:** Thaddaeus Muri, Madlaina Schöni, Felix W. A. Waibel, Dominique Altmann, Christina Sydler, Pascal R. Furrer, Francesca Napoli, İlker Uçkay

**Affiliations:** 1Technical Orthopedics and Neuro-Orthopedics Team, Department of Orthopedic Surgery, Balgrist University Hospital, Forchstrasse 340, 8008 Zurich, Switzerland; thaddaeus.muri@balgrist.ch (T.M.); madlaina.schoeni@balgrist.ch (M.S.); felix.waibel@balgrist.ch (F.W.A.W.); christina.sydler@balgrist.ch (C.S.); pascal.furrer@balgrist.ch (P.R.F.); francesca.napoli@balgrist.ch (F.N.); 2Infectiology, Balgrist University Hospital, 8008 Zurich, Switzerland; 3Unit for Applied and Clinical Research, Balgrist University Hospital, University of Zurich, 8008 Zurich, Switzerland

**Keywords:** diabetic foot infection, orthopedic surgery, preoperative antibiotics, postoperative antibiotics, antibiotic duration, outcome, clinical failure, infection recurrence

## Abstract

Many patients with community-acquired diabetic foot infections (DFI) receive systemic (empirical) antibiotic treatments before surgical interventions, often by the general practitioner. Sometimes, hospital clinicians prescribe them before surgery to reduce the infection inoculum and preserve soft tissue for immediate wound closure in case of residual infection after surgery. In contrast, the international guidelines (IWGDF) advocate against presurgical antibiotic use in routine situations without severe progredient soft tissue infections and/or sepsis. We run several retrospective and prospective cohorts of DFI and retrospectively analyze the influence of presurgical antibiotic therapy (as binary (yes/no) or continuous (in days) variables) on failures after a combined surgical and medical treatment. In our large database, the presence, choice, administration routes, or duration of preoperative antibiotic therapy did not improve the postoperative outcomes of operated diabetic foot infections or prevent their failures. In turn, this lack of influence leaves space for enhanced antibiotic stewardship in the management of DFI.

## 1. Introduction

The number of diabetic foot infections (DFIs) is increasing worldwide as they constitute a huge burden for patients, healthcare systems, patients, and clinicians [1,2,3,4]. The correct antibiotic regime and its timing for each of the various subsets of DFIs are an ever-evolving field and subject of current research. Many patients with community-acquired diabetic foot infections (DFI) are under systemic (empirical) antibiotic treatments before surgical interventions for multiple reasons [5,6]. Often, the general practitioner starts an empirical antibiotic therapy to avoid hospitalization and/or surgery [7]. Sometimes, hospital clinicians also prescribe systemic empirical antibiotic agents upon admission, even if the surgery is already planned. Possible reasons for this non-adherence to widespread common surgical practice and expert international guidance [8] include a conviction to reduce the local bacterial load, protect the surrounding soft tissue in favor of better wound healing (primary closure) after surgery [5], or start proactively (for the sake of patients’ comfort) an intervention motivated by the fact that surgery cannot take place immediately.

However, this eventually hasty practice may have its downsides. Presurgical antibiotic treatment might expose the patients to antibiotic-related adverse events [9,10], which are frequent among frail diabetic foot patients, and compromise the yield of intraoperative microbiological sampling [11,12], or alter the microbiological results shifting toward more resistant pathogens via “selection” under ongoing therapeutic antibiotic use [13]. The reasons for delaying surgery in the hospital might be equally multiple, i.e., waiting for the waning of the anticoagulation, preoperative revascularization or dialysis, or simply a question of the timely availability of operation slots and/or experienced surgeons.

Therefore, the indication for routinely (or preventively) starting the antibiotics might preoperatively not be evidence-based and work against the principle of antibiotic stewardship in the field of DFI [2,14,15]. For years, the International Working Group for the Diabetic Foot (IWGDF) has repetitively advocated against presurgical empirical antimicrobial use in routine situations without severe progredient soft tissue infections and/or sepsis [8,16]. To the best of our knowledge, in the adult DFI population, the influence of the modalities and duration of preoperative antibiotic use on postoperative outcomes has not been evaluated in a scientific way, which has been lamented in the last IWGDF guidance on infection [8,16]. Such a study question requires a case-control design, as prospective-randomized trials with fixed durations of antibiotic prescription [17] cannot adequately assess it without major ethical concerns. In this study, we retrospectively analyze the influence of presurgical antibiotic treatment (as binary (yes/no) or continuous (in days) variables) on failures after a combined surgical and medical treatment for operated DFIs.

## 2. Results

### 2.1. General Results

We included 1235 moderate to severe DFI cases: 266 females (22%); 84% type II diabetes; 927 (75%) under insulin medication; 700 (57%) active smokers with a median of 35 pack-years (interquartile range (IQR), 20–60 p-y); 265 patients (21%) with heart and 624 persons (51%) with chronic renal insufficiency, of which 105 (9%) in renal dialysis; 29 (2%) with solid organ transplantations, 302 (24%) with high alcohol consumption, 76 (6%) under immune-suppressive therapy, among which 20 episodes under oncologic chemotherapy.

In both groups with and without preoperative antibiotic use, most patients were male (78%). Insulin use was more common in the preoperative antibiotic group (*p* = 0.008). There were no other significant differences besides the fact that one group received presurgical antibiotics and the other did not (Table 1). Preoperative antibiotic therapy was prescribed either by the referring physician or by our unit. In multiple cases, antibiotic therapy was prescribed for a non-DFI-related infection, e.g., a urogenital infection, prior to admission to our hospital. Due to the heterogeneity of available written information and the frequent lack of justification of preoperative antibiotics in the medical files and admission letters, we were unable to explore any rationale for antibiotic prescription in those cases who did not have concomitant infections. It was a multifaceted decision that might have depended on either the referring physician’s knowledge of antibiotic stewardship or, in case of in-hospital prescription, the individual surgeon and his/her medical habitude, the presence of erysipelas, the patient’s co-morbidities, fever and/or necrosis. At database closure, 337 (27%) patients had died during the follow-up for various reasons.

### 2.2. DFI Pathogens

We found a total of 119 different microbiological constellations, with the majority (64%) representing monomicrobial DFIs (Institute for Medical Microbiology, University of Zurich). The five most frequently retrieved pathogen groups were coagulase-negative staphylococci, *Staphylococcus aureus*, enterococci, streptococci, and *Pseudomonas* spp.

### 2.3. Therapeutic Interventions

The median number of surgical interventions (debridement) for infection was one (IQR, 1–2 debridement per episode), the median duration of presurgical antibiotic treatment was 13 days (IQR, 5–27 d), and the median duration of postsurgical (mostly targeted) antibiotic treatment was 21 days (IQR, 12–41 d). Overall, 912 DFI episodes (74%) revealed a presurgical antibiotic prescription within two weeks before surgery, prescribed by the general practitioner in more than 90% of cases. In 47 DFI cases, the median antibiotic-free “window” before surgery was four consecutive days (IQR, 2–11 days). We prescribed 71 different postsurgical antibiotic regimens based on the pathogens, susceptibility profiles, drug interactions, the clinicians’ personal choices, and the other existing co-morbidities of the patient. In one-third of cases, the clinicians had started with several antibiotic agents simultaneously. Among all choices, the initial (empirical) postsurgical therapy for the 386 DFI episodes was broad-spectrum agents or a combination of agents providing broad-spectrum coverage. Among 908 (74%) episodes with (occasionally) symptomatic peripheral arterial diseases (Table 1), 662 (54%) were revascularized in the limb/leg, and 44 (4%) underwent a cardiac bypass before or during the study period.

### 2.4. Outcomes

Overall, 299 (24%) clinically failed within one year for various reasons (ischemia, new infection, new ulcer), and in 60 (5%) cases, we witnessed a microbiological recurrence of infection involving the same pathogens as in the index episode (Table 1). In retrospective group comparison, the presence of presurgical antibiotic therapy was moderately associated with overall “failure” (676/935 vs. 235/299 episodes, Pearson χ^2^-test, *p* = 0.03), but not with the outcome “microbiologically-identical recurrences” (856/1168 vs. 49/60, *p* = 0.15). In separate multivariate logistic regression analyses, the duration of presurgical treatment (odds ratio 1.0, 95% confidence interval 0.99–1.01) and the duration of postoperative therapy (OR 1.0, 95%CI 0.99–1.01) were completely unrelated to “failure” or “microbiological recurrence” (OR 1.0, 1.0–1.0 and OR 1.0, 1.0–1.0), respectively (insignificant goodness-of-fit tests in all analyses). The odds ratio and the confidence intervals oscillated very narrowly around number 1, highlighting that the pre- and post-operative antibiotic durations had only a very limited effect, or not at all, on the outcome “failure” (Table 2). This is in contrast with ischemia, which was much more determinant of “failure”.

Our database was too heterogenous to dissect if any particular antimicrobial agent, in monotherapy or as a combined treatment, would change the general findings. Statistically speaking, we cannot control for the large case mix with its different infection strata [18], patients’ co-morbidities [19], infection localizations [20], causative pathogens [21,22], antimicrobial agents and doses [23], surgical techniques [24], drug interactions with myriads of co-medications [25], and important ischemia parameters [19]. Nonetheless, at a glance at the composite database, we failed to detect any specific presurgical antibiotic patterns, as per the observational lecture, which would have demonstrated a striking difference.

## 3. Discussion

According to our single-center case-control study, the presence, choice, administration routes, or duration of preoperative antibiotic therapy did not improve the postoperative outcomes of operated DFI or prevent their failures. In retrospective group comparison, the presence of this presurgical antibiotic therapy was moderately associated with overall postsurgical failure (676/935 vs. 235/299 episodes), which encompasses all therapeutic failures such as wound breakdowns, but not with microbiologically-identical recurrences (856/1168 vs. 49/60 DFI cases). In separate multivariate logistic regression analyses, the duration of presurgical treatment (odds ratio 1.0, 95% confidence interval 0.99–1.01) and the postoperative therapy (OR 1.0, 95%CI 0.99–1.01) were entirely unrelated to “clinical failure” or “microbiological recurrence.” There is no benefit of preventive (or premature) antibiotic use preoperatively in clinically stable infections, not even in ischemic and/or necrotic tissues. In contrast, and very importantly, our findings must not be confounded with empirical antibiotics starting in rapidly spreading (bacteremic) infections. The distinction between both entities is a clinical decision (according to SIRS/sepsis criteria) because the routine serum inflammatory markers, such as the C-reactive protein or the (differentiated) leukocyte count, are not determinants of the final outcomes [7,16].

Surprisingly, a high number (74%) of adult DFI patients received presurgical antibiotic treatment, even though this is largely discouraged by international guidance [8,16]. Although the data of our study do not identify why these guidelines were not followed, they coincide with recent research [6]. In a previous work in Geneva by the leading author, 43% of all infected orthopedic patients came to the hospital with an established empirical antibiotic therapy. In a recent Turkish evaluation, this proportion was 74% among diabetic patients admitted for transtibial amputation [26]. Of note, both comparative groups (with and without preoperative antibiotics) were equally distributed in terms of patients’ characteristics, with the exception of more insulin use in pretreated patients, for which the reasons remained unclear. We speculate that the patients under insulin yield a long-lasting relationship with their general practitioner, who was more likely to start an antimicrobial therapy than patients with newer onsets of diabetes mellitus. Obviously, therapeutic traditions are more obdurate and stubborn than expert guidelines and take time to change, especially when considering frail DFI patients in the context of multidisciplinary management in and outside of the hospital.

Our findings are not surprising. The administration of antibiotics is certainly a game changer and a powerful therapeutic tool in acute soft tissue DFIs. In contrast, in operated, chronic DFI cases marked by ischemia, ulcerations, and osteitis, the power of antibiotics is relatively diminished in favor of other co-interventions such as surgery, revascularization, or off-loading [1,2,3,4,16]. We are unaware of any scientific publication in which the duration of the pre- or postoperative antibiotic therapies would significantly determine the fate of operated DFIs. Indeed, in several different papers, neither the duration of postoperative systemic antibiotic prescription, their administration form (oral, intravenous, or local), nor the choice of the corresponding agent [7] (molecule, targeted vs. empirical, broad vs. narrow-spectrum) determined outcome for the majority of our multimorbid patients. Treating with an (over) aggressive antibiotic regimen is a problem not only for DFI but also for other populations affected by infections. Narrowing the spectrum of antibiotic regimens and limiting the duration of antibiotic use are some of the most important goals within the antibiotic stewardship projects. In an earlier study, we showed that a narrow-spectrum antibiotic regimen based on local epidemiology is not inferior to a broad-spectrum antibiotic regimen in DFI [7]. Similar work has been done in different infection populations, such as respiratory infections, with the same effect of promoting a narrow-spectrum antibiotic treatment and a de-escalation of the therapy whenever feasible [27,28,29].

The strengths of our study are a large database, a long minimal follow-up of six to twelve months, a single-center evaluation, and a multidisciplinary team experienced in DFIs. To our knowledge, this study is the first to assess the effect of presurgical antibiotic treatment on the postoperative outcome in community-acquired DFIs.

Our study also has obvious limitations. Besides its retrospective design and the corresponding “confounding by indication”, our cohort primarily consists of moderate to severe DFI cases that underwent surgical procedures. Therefore, we cannot address the relevance of mild DFI episodes, which form the bulk of the DFI in the general practitioner’s office. Secondly, by concentrating on our study question, we chose not to analyze the impact of other variables, such as antibiotic-related costs, adverse events of antimicrobial therapies, glycemic control, or malnutrition [7]. Concerning the possible adverse events, it is extremely difficult to distinguish the antimicrobial cause from other medications and between the presurgical and the immediate postsurgical period [24]. Thirdly, we live in a resource-rich, small city with a low endemicity of multi-resistant pathogens. Our findings could be different in other settings with difficulties accessing costly intravenous medication [30,31] or high endemicity of community-acquired, (gram-negative) multi-resistant pathogens [21,23,32,33]. Fourthly, we ignore the exact reason for a preoperative antibiotic prescription. Our mixed author group treats patients from the time-point of hospitalization. We made efforts to withhold antibiotics until the intraoperative tissue samples were collected. However, at minimum, 95% of the preoperative antibiotic prescriptions were initiated by the general practitioners (GP or other colleagues in other hospitals) who diagnosed infection on clinical grounds and started the empirical treatment despite the recommendation to withhold until sampling, regardless of the hemodynamic “stability” of the patient. In that sense, we can say that all cases had a certain amount of visible soft tissue infection because the GP started in case of a visible soft tissue infection. Hence, we did not witness any cases of preoperative antibiotic use when there was only chronic osteomyelitis without any visual soft tissue involvement.

However, what we cannot reconstruct are the detailed reasons. We ignore which proportion of preoperative treatment episodes were mostly due to fear of spreading before surgery, misinterpretation of ischemia, ignorance, the demand of patients and families, fear of legal consequences, or an attempt to resolve the problem conservatively without surgical debridement. Likewise, we ignore the degree of the clinical experience of the various GPs or in the emergency settings of the peripheral hospitals. For all these reasons, and in the presence of osteomyelitis without substantial soft tissue involvement, our study question becomes obsolete in episodes with only a chronic ulcerated diabetic toe osteomyelitis (DFO) without soft tissue involvement.

Whenever possible, a targeted antibiotic therapy guided by a prior bone biopsy should be chosen, whether a surgical or conservative treatment of sole DFO episode is performed. In addition, the treatment modalities in such cases are more fixed according to expert recommendations [16,34].

## 4. Materials and Methods

### 4.1. Setting and Database

The Balgrist University Hospital is a tertiary center in Zurich and has a specialized unit for DFI. We run several prospective cohorts and randomized trials on the management of DFIs [7,13,17,25]. For this study, we performed a case-control study that was embedded in a composite cohort of adult DFI patients treated at the Balgrist between 1 January 2014 and 29 February 2024, with a minimal follow-up of six months following the end of the last treatment. Database closure occurred on 15 September 2024. An individual patient could participate several times in this study, provided he/she revealed new DFI episodes at different localizations on both feet. We published the preliminary results in 2024 as a poster at the Swiss national congresses for Orthopedic Surgery [35] and Infectious Diseases [36].

### 4.2. Study Definitions

We defined “Remission” as an absence of any clinical, imaging, or laboratory findings suggesting the failure of treatment after a minimal follow-up of six months after the end of therapy. We defined “Clinical Failure” as the need for any surgical revision or the occurrence of a new DFI episode needing prolonged antibiotic treatment at the same anatomical site of the foot up to one year after index surgery. Hence, a clinical failure could have many causes, such as new ischemia, late development of a diabetic foot ulcer, a bone fracture, or a new infection episode. We defined “Microbiological Recurrence” as an infection relapse specifically due to the same pathogen(s) identified at the index (prior) infection. As we lacked microbiological typing of recurrent bacterial strains, we relied on the species identification together with the corresponding antibiotic susceptibility testing results.

### 4.3. Statistical Analyses

For group comparisons, we used the Pearson-χ^2^-square test for categorical variables and the Wilcoxon rank-sum test for non-parametric continuous variables. We performed separate risk factor analyses for the endpoints “Clinical Failure” and “Microbiological Recurrence,” using unconditional uni- and multivariate logistic regression analyses. For both outcomes, the final models consisted of the following variables: antibiotic durations pre- and postoperatively, biological sex, revascularization, and the number of surgical debridements for the initial (index) DFI. We used SPSS^™^ (IBM; Vers. 26) and STATA^™^ software (Vers. 15, College Station, TX, USA) and considered *p*-values ≤ 0.05 (two-tailed) as significant.

## 5. Conclusions

Our data provides more arguments for withholding unnecessary antibiotic prescriptions before surgery and enables physicians and surgeons who are willing to do so to comply with the worldwide efforts of antibiotic stewardship in the preoperative setting of DFIs. Overprescribing antibiotics for mild or moderate DFIs very presumably contributes to the development of MDR bacteria, hampers the diagnostic yield of intraoperative microbiological samples, and probably leads to a broader-spectrum postoperative antibiotic therapy due to the selection of (more resistant) pathogens. As presurgical antibiotic coverage does not alter the postoperative fate and therapy in the slightest amount, we strongly encourage abandoning this widespread practice for reasons of convenience or prevention and reserving it only in case of proven rapidly spreading soft-tissue infections and/or sepsis.

## Figures and Tables

**Table 1 antibiotics-13-01136-t001:** Group comparison between patients with and without presurgical antibiotic treatment.

Preoperative Antibiotics?		No Antibiotics		Preoperative Antibiotics		
		Number	(%)	Number	(%)	*p*-Value
Female gender		69	21.5%	196	21.5%	0.999
Median Age (years)		67.36		66.26		0.075
Type Diabetes	Type I	35	10.9%	135	14.9%	
	Type II	278	86.9%	763	83.9%	0.204
Insulin use	Yes	223	69.7%	703	77.1%	0.008
Renal insufficiency	Yes	163	51.1%	460	50.8%	0.934
Renal Transplantation	Yes	8	2.5%	16	1.8%	0.403
Peripheral Artery Disease	No	87	27.2%	239	26.2%	0.787
	Stadium I	63	19.7%	186	20.4%	
	Stadium II	92	28.8%	232	25.4%	
	Stadium III	2	0.6%	9	1.0%	
	Stadium IV	76	23.8%	246	27%	
Coronary heart disease	Yes	137	42.8%	422	46.3%	0.285
Cardiac insufficiency	Yes	66	20.6%	198	21.7%	0.677
Enhanced immune-suppression *	Yes	20	6.2%	56	6.1%	0.954
Active smoking	Yes	189	59.4%	510	56.2%	0.321
Alcoholism	Yes	75	23.6%	227	25.1%	0.601

* Enhanced immune suppression = Immune suppression beyond diabetes: dialysis, liver cirrhosis CHILD B or C, organ transplantation, immune-suppressive medication.

**Table 2 antibiotics-13-01136-t002:** Unconditional logistic regression analyses (separate outcomes for clinical failures and microbiological recurrences) (results expressed as continuous variables and as odds ratios (OR) and 95% confidence intervals (CI).

	*Clinical Failures (n = 299)*		*Microbiological Recurrence (n = 60)*
	Univariate OR (95% CI)	Multivariate OR (95% CI)	Multivariate HR (95% CI)
Female sex	* **1.5 (1.1–2.2)** *	1.3 (0.8–2.4)	0.5 (0.2–1.4)
Duration pre-operative antibiotics	1.0 (0.99–1.01)	1.0 (0.99–1.01)	1.0 (0.99–1.01)
Duration post-operative antibiotics	1.0 (1.00–1.01)	1.0 (1.00–1.01)	1.0 (0.99–1.01)
Number of surgical debridement	0.9 (0.8–1.1)	1.0 (0.9–1.1)	1.1 (0.9–1.3)
Clinical need for revascularisation	* **1.5 (1.2–2.0)** *	* **1.9 (1.2–2.8)** *	0.7 (0.4–1.4)

Clinical failure = Surgical revision for any reason, e.g., infection recurrence, new infection, ischemia, skin breakdown, hematoma. Microbiological recurrence = Recurrence of the index infection with the same pathogens. Significant results are in ***bold and italics***.

## Data Availability

Key data may be available in an anonymized form upon scientific request to the corresponding author.

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
