# Peer review of "Preoperative Antibiotic Administration Does Not Improve the Outcomes of Operated Diabetic Foot Infections†"

_antibiotics, 2024, doi:10.3390/antibiotics13121136_

Round 1

Reviewer 1 Report

Comments and Suggestions for Authors

The authors retrospectively analyzed 1,235 moderate to severe DFI cases to investigate whether preoperative antibiotic treatment impacts the outcomes of operated diabetic foot infections. The authors found that based on their data collected from a single center in Zurich, antibiotic administration does not change the clinical failure status or microbiological recurrence. The authors call for enhanced antibiotic stewardship in the management of DFI.

Below are my specific questions:

1. As the authors speculated, the usage of antibiotics tends to be partially due to obdurate and stubbornness compared to expert guidelines. What's the percentage of cases with visible soft-tissue infection that require antibiotic treatment? To what extent does this subset of the patient population benefit from presurgical antibiotic treatment?

2. What's the time point to determine Clinical Failure and Microbiological Recurrence? What's the rationale for setting the time point? Would difference in follow-up time confuse the result?

Author Response

Reviewer 1:

The authors retrospectively analysed 1,235 moderate to severe DFI cases to investigate whether preoperative antibiotic treatment impacts the outcomes of operated diabetic foot infections. The authors found that based on their data collected from a single centre in Zurich, antibiotic administration does not change the clinical failure status or microbiological recurrence. The authors call for enhanced antibiotic stewardship in the management of DFI.

Below are my specific questions:

  1. As the authors speculated, the usage of antibiotics tends to be partially due to obdurate and stubbornness compared to expert guidelines. What's the percentage of cases with visible soft-tissue infection that require antibiotic treatment? To what extent does this subset of the patient population benefit from presurgical antibiotic treatment?

Answer: Yes. We now explain better in the "Limitations" (lines 210-228).

Our mixed author group treats patients from the time-point of hospitalization. By ourselves, we made efforts to withhold antibiotics until the intraoperative tissue samples. However, at minimum 95% of the preoperative antibiotic prescription was initiated by the general practitioners (GP; or other colleagues in other hospitals) who diagnosed infection on clinical grounds and started the empirical treatment despite the recommendation to withhold until sampling; regardless of the hemodynamic "stability" of the patient. In that sense, we can say that all cases had a certain amount of visible soft tissue infection because the GP start in case of a visible soft tissue infection. Hence, we did not witness any cases with preoperative antibiotic use when there was only chronic osteomyelitis without any visual soft tissue involvement.

However, what we cannot reconstruct are the detailed reasons. We ignore which proportion of preoperative treatment episodes were mostly due to fear of spreading before surgery, misinterpretation of ischemia, ignorance, the demand of patients and families, fear of legal consequences, or an attempt to resolve the problem conservatively without surgical debridement. Likewise, we ignore the degree of the clinical experience of the various GPs or in the emergency settings of the peripheral hospitals.

  1. What's the time point to determine Clinical Failure and Microbiological Recurrence? What's the rationale for setting the time point? Would difference in follow-up time confuse the result?

Answer: Yes. We defined Remission as the absence of any finding (clinical, imaging, laboratory) suggesting failure of treatment after a minimal follow up of six months (lines 260-261). Clinical Failure was the need for any surgical revision, or the occurrence of a new DFI episode needing prolonged antibiotic treatment, at the same anatomical site of the foot up to one year after index surgery (lines 262-263).

The duration of follow-up would only shift the proportion of microbiologically-identical relapses versus new infections at the same localization. The longer you follow, the more is the chance of a new infection in case of "Failures". We must not confound with the traditional two-years follow-ups in treated prosthetic joint infections, which is due to the implant-related nature of infection. PJI’s and DSFIs are totally different in epidemiology, outcome and conceptual treatment.

In our study, we use very long follow-up times for diabetic foot infections. In the literature, DFI relapse within some weeks or months. Episodes after longer follow-up periods are practically univocally new infections, even if the species of the causative pathogens might be identical. International experts acknowledge this. In the most recent IWGDF guidelines in 2023 reduce the follow-up to 6 months. Microsoft Word - 04 - Infection Guideline.docx. Recommendation Nr. 18.

We fulfil these new follow-up requirements and will not go into details. We now briefly mention our long follow-up period in lines 194-195.

Reviewer 2 Report

Comments and Suggestions for Authors

Thank you very much for performing this research and sharing your valuable data with the scientific community.

In the introduction section, the authors have clearly demonstrated the importance of their research and provided background information citing available data in the field. DFI is a challenging entity in medical practice. Moreover, empirical antibiotics are prescribed even without a definitive indication. The presented data have clearly showed this challenge in daily practice.

I am more than satisfied with the presented data, presentation method and discussion regarding this submission and believe that this data will be a well-fit for Antibiotics. 

However, as the authors possess such an enormous data base, they may consider implementing an antimicrobial coverage score (PMID 34805436) and evaluate its effect on the clinical outcome/failure in DFI.

Author Response

Reviewer 2:

Thank you very much for performing this research and sharing your valuable data with the scientific community.

In the introduction section, the authors have clearly demonstrated the importance of their research and provided background information citing available data in the field. DFI is a challenging entity in medical practice. Moreover, empirical antibiotics are prescribed even without a definitive indication. The presented data have clearly showed this challenge in daily practice. I am more than satisfied with the presented data, presentation method and discussion regarding this submission and believe that this data will be a well-fit for antibiotics

Answer: We thank you very much.

However, as the authors possess such an enormous data base, they may consider implementing an antimicrobial coverage score (PMID 34805436) and evaluate its effect on the clinical outcome/failure in DFI.

Answer: This is a very pertinent remark, but unfortunately not the primary objective of this paper. Our actual paper investigates the influence of preoperative antibiotic (durations) on the postoperative outcomes, and not the completeness of the (empirical) coverage.

We have already addressed this coverage issue in another recent publication (reference 7). In the publication of Nieuwland et al., a similar approach was chosen with the question of the completeness of antibiotic coverage. Specifically, in that retrospective study of 761 diabetic foot infections episodes the antibiotic regime was stratified into adequate empirical therapy, culture guided therapy and empirical inadequate therapy. It was also assessed if an empirical broad versus a narrow-spectrum therapy did alter the outcome. According to our data the microbiological adequacy of the initial antibiotic treatment regimen did not alter the therapeutic outcomes. We concluded that clinicians should follow the common antibiotic stewardship approach of starting with a narrow-spectrum antibiotic regimen based on the local epidemiology instead to a broad-spectrum regimen with than a de-escalation in the course of treatment [7].

Due to our already published work and in order to keep this actual one as precise as possible, we now only mention the coverage issue briefly in our revised version (Discussion, line 186-193) by introducing the recommended paper (PMID 34805436) as a new reference 32 (line 193).